# Flexible NAD^+^ Binding in Deoxyhypusine Synthase Reflects the Dynamic Hypusine Modification of Translation Factor IF5A

**DOI:** 10.3390/ijms21155509

**Published:** 2020-07-31

**Authors:** Meirong Chen, Zuoqi Gai, Chiaki Okada, Yuxin Ye, Jian Yu, Min Yao

**Affiliations:** 1Faculty of Advanced Life Science, Hokkaido University, Sapporo 060-0810, Japan; chenmr1987@163.com (M.C.); gaizuoqi@126.com (Z.G.); chiaki@hoku-iryo-u.ac.jp (C.O.); yeyx@pkusz.edu.cn (Y.Y.); yu@castor.sci.hokudai.ac.jp (J.Y.); 2College of Food Science and Technology, Nanjing Agricultural University, Nanjing 210095, China

**Keywords:** IF5A, translation factor, hypusine modification, deoxyhypusine synthase, structure, NAD^+^

## Abstract

The eukaryotic and archaeal translation factor IF5A requires a post-translational hypusine modification, which is catalyzed by deoxyhypusine synthase (DHS) at a single lysine residue of IF5A with NAD^+^ and spermidine as cofactors, followed by hydroxylation to form hypusine. While human DHS catalyzed reactions have been well characterized, the mechanism of the hypusination of archaeal IF5A by DHS is not clear. Here we report a DHS structure from *Pyrococcus horikoshii OT3* (*Pho*DHS) at 2.2 Å resolution. The structure reveals two states in a single functional unit (tetramer): two NAD^+^-bound monomers with the NAD^+^ and spermidine binding sites observed in multi-conformations (closed and open), and two NAD^+^-free monomers. The dynamic loop region V288–P299, in the vicinity of the active site, adopts different positions in the closed and open conformations and is disordered when NAD^+^ is absent. Combined with NAD^+^ binding analysis, it is clear that *Pho*DHS can exist in three states: apo, *Pho*DHS-2 equiv NAD^+^, and *Pho*DHS-4 equiv NAD^+^, which are affected by the NAD^+^ concentration. Our results demonstrate the dynamic structure of *Pho*DHS at the NAD^+^ and spermidine binding site, with conformational changes that may be the response to the local NAD^+^ concentration, and thus fine-tune the regulation of the translation process via the hypusine modification of IF5A.

## 1. Introduction

Hypusine (*N*-(4-amino-2-hydroxybutyl) lysine), an unusual amino acid, is formed by the post-translational modification of a lysine residue with the addition of the aminobutyl moiety from the polyamine spermidine. The hypusine modification uniquely occurs in eukaryotic translation factor 5A (eIF5A) precursor proteins [1]. Although eIF5A was originally designed as a translation initial factor, it has been shown that eIF5A is more directly related to elongation rather than the initiation step. The hypusine-modification of eIF5A is involved in the mRNA transportation and translation elongation on the ribosome [2,3]. The binding of eIF5A to translating ribosomes in a hypusine-dependent manner indicates modifications of the specific binding to the translational machinery [4,5], suggesting the important role of eIF5A receiving hypusine modifications in translation processes, which was supported by evidence that the hypusine modification of eIF5A is vital for the growth and survival of eukaryotes, from yeast to mammals [6,7,8]. Previous studies also reported that the hypusine residue is essential for the homodimerization of eIF5A and affects its subcellular location [1,2,9,10].

The hypusine modification of eIF5A is catalyzed by deoxyhypusine synthase (DHS), followed by the hydroxylation of deoxyhypusine by deoxyhypusine hydroxylase (DOHH) [1,11]. DHS catalyzes a NAD^+^-dependent reaction that synthesizes deoxyhypusine (*N*-(4-aminobutyl) lysine) by transferring the butylamine moiety of spermidine to a specific lysine residue of eIF5A [12]. Then, deoxyhypusine is hydroxylated by DOHH to form hypusine [13,14,15]. Like eIF5A, DHS is an essential gene, indicating the importance of hypusine modification [16]. Since eIF5A is associated with many diseases, such as diabetes, cancers, malaria, and HIV-1 infections [16], trials aimed at blocking the hypusine modification of eIF5A have been attempted. Specific inhibitors of DHS or DOHH have been reported to affect the arrest of cellular growth, and also alter the proliferation and differentiation of tumor cells [17,18,19]. An extensive biochemical characterization of DHS has been carried out, establishing it as a 40–43 kDa enzyme (the exact size being species-dependent) requiring NAD^+^ as a cofactor. The mechanism of human deoxyhypusine synthesis by DHS has been proposed as follows [20]: firstly, a hydrogen of spermidine is transferred to NAD^+^ to generate dehydrospermidine and NADH; the dehydrospermidine then links covalently to a lysine residue (K329 in human DHS) of DHS, forming an enzyme-imine intermediate; finally, DHS catalyzes the formation of deoxyhypusine through the conjugation of the butylamine moiety of dehydrospermidine to the ε-amino group of the lysine residue (K37) of the eIF5A precursor protein.

The reported crystal structures of human DHS show a homotetramer consisting of two tightly associated dimers with four active sites—two near each tight dimer interface [21,22]. Residues from both monomers of a dimer contribute to each of its active sites. In human DHS, four active sites accommodate four molecules of NAD^+^ (PDB:1DHS) [21,22]. Interestingly, the catalytic sites are blocked by a projecting N-terminal two-turn helix (described as a “ball and chain”) in a cross-monomer manner in the first crystal form of human DHS, obtained upon crystallization at high ionic strengths and a low pH. However, the catalytic site is not obstructed in a structure of human DHS in a ternary complex with NAD^+^ and GC7 (PDB:1RQD), an analog of spermidine [22]. In this ternary complex structure, GC7 is also located at the interface of the tight dimers and is separated from NAD^+^ by a loop structure (T307–A319) [22]. The absence of a blockage by the ball and chain structure might suggest that this dynamic part of the enzyme could be involved in the regulation of activity within the functional tetramer. However, this human DHS complex structure without any bound GC7 (PDB:1ROZ), determined upon crystallization at low ionic strengths and pH 8.0 (nearer physiological conditions), lacked the electron density for the N-terminal ball and chain region [22]. In recent research, it has been reported that the N-terminal ball and chain region is important for the assembly of a functional DHS tetramer [23]. This N-terminal region is, therefore, clearly highly dynamic, with the flexibility being for DHS function. Whether it is regulating the active site accessibility in a substrate-dependent manner, binding eIF5A, or conferring sensitivity to cellular conditions, remains unclear. Different from human DHS, the DHS from *Trypanosoma brucei* exists as a heterotetramer formed by two paralogs, DHSc and DHSp (dimer of heterodimer), with DHSc homologous to human DHS. Each heterodimer contains two NAD^+^ binding sites, one located in the functional catalytic site while the other is located in a remnant site lacking key catalytic residues [24].

Consistent with archaea sharing a requirement for hypusine for the function of translation factor aIF5A, both the modification [24] and DHS gene homologues [1] have been identified. Furthermore, a study has shown the arrest of archaeal growth in G1 upon treatment with an inhibitor of DHS [25], indicating that DHS is also essential in archaea. However, the mechanism of hypusination in archaea is unclear yet [26]. Homologues of the non-essential second enzyme required for hypusine formation, DOHH, have not been identified. While some archaea may function with a deoxyhypusine modification (rather than hypusine), species are known that form hypusine in the absence of oxygen, which implies a distinct mechanism of hypusine modification in archaea. The structure of an archaeal DHS could provide a better understanding of the process of hypusine modification of aIF5A.

In this study, we determined the crystal structure of DHS from *Pyrococcus horikoshii* OT3 (*Pho*DHS), with NAD^+^-bound, at 2.2 Å resolution. *Pho*DHS is a tetramer, each monomer sharing a similar fold with its human homologue (PDB ID:1DHS). Interestingly, unlike human and *T. brucei* DHS, *Pho*DHS contains only two bound NAD^+^ molecules and shows a difference in the NAD^+^-bound and NAD^+^-free monomeric conformations. In particular, NAD^+^-bound monomers in the *Pho*DHS tetramer display different conformations in the region of V288–P299 around the NAD^+^ binding site. The observations are suggestive of a distinct manner in which NAD^+^ binds to DHS that could be of relevance to the mechanism and regulation of enzyme activity. Having the states of NAD^+^-bound and NAD^+^-free also sheds light on the NAD^+^-dependent spermidine binding to DHS and raises a possibility that dynamic NAD^+^ binding in *Pho*DHS may fine-tune the translation activity of aIF5A via hypusine modification.

## 2. Results

### 2.1. The Overall Structure of PhoDHS

We determined the crystal structure of *Pho*DHS at 2.2 Å resolution by using a molecular replacement (Table 1). In an asymmetric unit, the structure of *Phs*DHS formed a tetramer in accordance with its known biological assembly [21,22], with unbuilt N-terminal 14 and C-terminal 6 of the total 342 residues in each monomer. Additionally, we could not build the parts of two loops in each monomer (E122-D125 and G306–K311 in molA, W290–K311 in molB, E122–V123 and W290–A310 in molC, and W121–K130 and E301–K311 in molD, respectively), because of the lack of electron density. Similar to DHS from other organisms, this tetramer is a dimer of dimers with a point group of D2, generated by two local 2-fold axes that are perpendicular to each other (Figure 1). The monomers of *Pho*DHS tetramers are more related to each other with a root mean squared deviation (r.m.s.d) between 0.544 Å and 0.606 Å for 276–284 Cα atoms. Based on the buried areas of intermolecular interactions, the tightly bound dimers are defined as molAB and molCD (buried areas > 2700 Å^2^), while the buried area is less than 1800 Å^2^ between molA and molD (or between molB and molC). *Pho*DHS is very similar to human DHS (PDB ID:1DHS), with a r.m.s.d of 1.195 Å for 289 Cα, and is also similar to *T. brucei* DHS (PDB:6DFT) [24] with a r.m.s.d of 1.39 Å for 275 Cα (Figure 1b). Differences include the N-terminal region (M1–P26) and an insertion (L77–L94) in human DHS. Unlike the inactive human DHS structure, wherein the N-terminal two-turn α helix (A9–L18) blocks the entrance tunnel to the spermidine active site [21,22], the N-terminal region (M1–I14) of *Pho*DHS is not built due to the poor electron density for this region, suggesting that it is highly flexible and different from human DHS, despite the crystallization conditions being similar to those of inactive human DHS [21]. Given that the active sites are accessible, we assume that the structure of *Pho*DHS obtained is illustrative of an active form.

Although the overall structures of DHS are similar, there is a significant structural difference in a loop region (V288–P299) that forms part of the active site of *Pho*DHS. This active site loop is flexible and presents itself in multi-conformations, conf1 and conf2 in molA and molD, respectively, which are likely to represent two snapshots of the active site (Appendix A). In the molB and molC of *Pho*DHS, the loop region is either disordered, or partially visible. Although only one of two conformations, conformation 1 or conformation 2 (conf1, conf2), was assigned to each molA and molD, a weak density of the alternative conformations could be observed, further suggesting the flexibility of the loop region in *Pho*DHS. In contrast, this active site loop is stable and present in just one conformation (conf2) in the previously published human and *T. brucei* DHS structures [24].

Another striking structural difference seen in *Pho*DHS is at its C-terminus. The C-terminus of *Pho*DHS (Gly337–Gln342, in all four chains) is invisible in the structure, which could be observed in human and *Trypanosoma brucei* DHS [24]. A sequence alignment shows that the primary structure of this region is not conserved in *Pho*DHS, human DHS, and *T. brucei* DHS (Figure 2). The short helix predicted for *Pho*DHS is longer in human DHS. The helix is visible in the human DHS structure, though the region retains some intrinsic flexibility, indicating conformational dynamics in the C-terminal region across species. Removal of five residues from the C-terminus in human DHS or 39 residues from the C-terminus of *S. cerevisiae* DHS impairs activity [26,27]. It is probable that the C-terminus of *Pho*DHS is also required for full function, and that the intrinsic flexibility of this region, rather than the conserved sequences, is likely important for the activity.

### 2.2. NAD^+^ Binding Manner of PhoDHS

Two bulky electron density blocks shaped as NAD^+^ were distinguished at the interface between molA and molD of the *Pho*DHS tetramer after several cycles of refinement and were later assigned as NAD^+^ (Figure 3a and Appendix A). Those NAD^+^ molecules were naturally captured during the expression of the enzyme in *Escherichia coli*, since no NAD^+^ was added during purification and crystallization processes.

Bound NAD^+^ is proximal to a loop (V288–P299 (hereafter named “the binding loop”)) that displays very different conformations in each monomer of the *Pho*DHS tetramer. Notably, in molA and molD, observed to bind NAD^+^, the binding loop adopts two different conformations (Figure 1a, right): in molA, the binding loop positions away from NAD^+^ (conf1), while in molD, the binding loop adopts another conformation to access NAD^+^ (conf2). In contrast, only the second conformation (conf2) of the binding loop was observed in the structures of human and *T. brucei* DHS (Figure 1b). In the molB and molC of the *Pho*DHS tetramer, where NAD^+^ is absent, the binding loop is disordered or only partially visible. We propose that NAD^+^ binding to *Pho*DHS induces the stabilization of the flexible loop structure, but that the alternative conformations indicate the action of the loop as a “lid.” In an open conformation (conf1), the active site is exposed to receive NAD^+^, and the lid is closed over NAD^+^ in conf2. This controlled switched direction of the binding loop in the presence of NAD^+^ may be important for the regulation of NAD^+^ and/or spermidine binding, the “lid” potentially opening and closing in response to the surrounding NAD^+^ content. In the absence of NAD^+^, the binding loop is presumably much more flexible in structure and thus invisible in molB and molC. Our *Pho*DHS structure is the first structure showing distinct differences in the NAD^+^-bound state of the DHS active site, differences additional to those in the NAD^+^-free state. This may be an additional pointer to a complex role for conformational dynamics in the function of DHS in the cell, potentially linked to the regulation of the translation process via a hypusine modification of IF5A.

The binding geometry of NAD^+^ in *Pho*DHS is similar to that in human DHS [21]. The two NAD^+^ molecules (NAD1 and NAD2) are embraced individually by molA and molD of *Pho*DHS in a sandwich binding manner and insert their ends into the pockets of molA, and molD with 2-fold symmetry (Figure 3b). The closest distance between NAD1 and NAD2 is 2.7 Å, where a hydrogen bond can form between the ribose hydroxyl group of each adenosine moiety (Figure 3a). Such interaction between NAD1 and NAD2 stabilizes their binding to DHS, which may explain why, when only two NAD^+^s are bound, these bind to molAD but not molAB. A cluster of residues from molA and molD compose a binding site for NAD^+^: NAD1 forms hydrogen bonds with Thr101, Ala102, Gly103, Glu107, Asp214, Ser262, Thr286, Ala287 and Asp320 from molA, and stacks with the imidazole ring of His266 from molD. Interestingly, the main chain groups of Leu294 and Ser295 of the binding loop from molD in conf2 are also involved in a hydrogen bonding network to NAD1. To further characterize the binding mode, we generate the two alternative conformations (conf1, conf2) in both molA and molD by the superimposition of the molA and molD, which shows that the binding loop in conf2 of molA is more tightly associated with the NAD2 (which shares a similarly bound geometry to NAD1) of molD than with NAD1. Except for Ser262, all residues that contact NAD^+^ are conserved (Figure 2). The crossover of interactions would suggest that molA and molD enhance NAD^+^ binding to each other, and there is likely to be a synergistic effect between the binding of NAD1 and NAD2, mediated by the flexible loop.

In the previously reported DHS structures [21,22,23], four NAD^+^ were accommodated individually in four active sites of the tetramer. In our *Pho*DHS structure, however, the occupation of only two active sites by NAD^+^ is observed for the first time. To further investigate the NAD^+^ binding state of the *Pho*DHS tetramer, we carried out a UV absorption scan using *Pho*DHS in solution (Figure 3c). There is an obvious difference in the absorption curves of *Pho*DHS alone and *Pho*DHS supplemented with NAD^+^. The match between the absorption scan for *Pho*DHS + 30 equiv NAD^+^ (molar ratio of *Pho*DHS:NAD^+^ = 1:30) and *Pho*DHS + two equiv NAD^+^ (molar ratio 1:2) indicates that *Pho*DHS can only accommodate two additional NAD^+^ even if excess NAD^+^ molecules are available, suggesting that there are two vacant sites in the purified *Pho*DHS, which become occupied in both conditions. The spectroscopy confirms that there are likely only two out of four binding sites that are occupied by NAD^+^ in purified *Pho*DHS in solution, validating the observations from crystallography. The fact that human DHS tetramers bind four NAD^+^ molecules at high concentrations of NAD^+^ [21,22] does not exclude the binding of only two NAD^+^ molecules when the NAD^+^ concentration is low. Conversely, our spectroscopy data show that *Pho*DHS can likely bind four NAD^+^ molecules when sufficient NAD^+^ is present. The lower binding ratio of NAD^+^ observed in *Pho*DHS could infer that DHS may also adopt distinct conformations for NAD^+^ binding in other species, probably regulated by the local concentration of cellular NAD^+^, which may offer a control mechanism for IF5A modification and, therefore, activity in cell processes.

### 2.3. Dynamic Conformation of the Region for Both NAD^+^ and Spermidine Binding

As described above, the comparison of *Pho*DHS monomers reveals the conformational dynamics near the active site. The region V288–K311, located in a valley of the DHS tetramer center, shows either free mobility (and therefore disorder) in the absence of bound NAD^+^ or two distinct conformations in its presence. In all four monomers, the electron density was lacking for G306–K311, which includes the catalytic Lys307, the imine bond acceptor with spermidine, and thus the aminobutyl donor to aIF5A. In a crystal structure of a human DHS ternary complex, this region is visualized and interacts with NAD^+^ and a spermidine analog, GC7 [22]. In a recent published paper, it was suggested that no conformational changes occur in the DHS structure upon binding with spermidine [23]. Our identification of the binding loop region V288–P299, acting as a lid over the active site to give open (conf1) and closed (conf2) forms, is indicative of a significant conformational change, at least with respect to NAD binding (Figure 3d). Upon the superposition of the *Pho*DHS and human DHS (complexed with GC7) structures, we found that GC7 has very close access (about 4.4 Å) to the V288–P299 loop in conf2, but is substantially far away (about 8.2 Å) in conf1 (Figure 3e). Thus, both NAD^+^ and spermidine can interact with the binding loop in conf2 but not in conf1, suggesting that the flexible binding loop plays a role in accepting substrates, even when NAD^+^ has bound. We can, therefore, consider three conformational states of this loop in relation to its function (Figure 4): (I) apo form: the disordered structure observed in molB and molC of *Pho*DHS, prior to NAD^+^ binding; (II) open form (conf1), the lid is away from the bound NAD^+^ (observed in molA) in *Pho*DHS, and is awaiting substrate binding; (III) reaction/closed form (conf2), the lid closes to contact the bound NAD^+^ (observed in molD) and also spermidine (as indicated in human DHS). Since *Pho*DHS interacts with NAD^+^, spermidine, and IF5A via this region, the conformational dynamics are perhaps an unsurprising potential contributor to the achievement of the different conformations for binding different partners. It is also possible, as stated earlier, that the conformational dynamics are a mechanism by which DHS can respond to the concentration of NAD^+^ and fine-tune the regulation of translation in organisms via the hypusine modification of IF5A.

## 3. Discussion

Taking the results as a whole, we propose that the binding of NAD^+^ to *Pho*DHS is regulated by the NAD^+^ concentration, which might itself be linked to the requirement for IF5A modification and the subsequent participation in translation. One *Pho*DHS tetramer binds two NAD^+^s at a low concentration of NAD^+^, but may be able to bind two additional NAD^+^s when supplemented with higher NAD^+^ concentrations. This new model for NAD^+^ binding by DHS is shown in Figure 4 with *Pho*DHS existing in three states: no NAD^+^, one *Pho*DHS tetramer binding two NAD^+^s on one side (molAD or molBC), and one *Pho*DHS tetramer binding four NAD^+^s. Our structural data definitively demonstrate the different conformations of loop V288–P299 in each monomer of *Pho*DHS, with the significant structural stabilization induced by NAD^+^ binding, which, with the help of a comparison to the human DHS:GC7 structure, is likely stabilized into the final closed conformation (conf2) upon binding spermidine. Indeed, the different conformations of the active site “lid,” or binding loop, observed in our structure are likely to reflect the absence of stabilizing interactions that spermidine binding would provide. Our structure provides a potential explanation for the NAD^+^-dependent binding of spermidine to DHS [29]. The conformation diversity observed in this region in our structure is compatible with an independence of NAD^+^ binding to each monomer, suggesting that monomers of DHS may behave equally whether in a tetramer, dimer or heterodimers. Strikingly, our structure presents a simultaneous combination of three different conformations at the spermidine binding site that are reflective of a three-state conformational regulation of DHS by NAD^+^: the apo form, the open form, and the closed/reaction form. The structure thus highlights the role of conformational dynamics in the regulation of DHS and hypusine modification of aIF5A.

To further address the relationship between the conformational changes of region V288–P299 observed here and the binding of spermidine and IF5A to generate hypusinated IF5A, the complex structure of IF5A-DHS is an important future goal.

## 4. Materials and Methods

### 4.1. Expression and Purification of PhoDHS

The gene encoding DHS (1–342 aa, 39 kDa) from *P. horikoshii OT3* (UniProtKB-O50105) was cloned into a modified pET26b expression vector with a hexahistidine (His_6_) tag and a TEV cleavage sequence (Glu-Asn-Leu-Tyr-Phe-Gln-Gly) at the N-terminus. For overexpression, the recombinant plasmid containing *Pho*DHS was transformed into *E. coli* strain B834 (DE3) pRARE2 by electroporation. The transformants were inoculated into Luria–Bertani (LB) containing 25 μg/mL kanamycin and 34 μg/mL chloramphenicol and cultivated at 30 °C until the optical density at 600 nm (OD_600_) reached about 0.6. After adding isopropyl β-d-1-thiogalactopyranoside (IPTG) to a final concentration of 1 mM, the cells were grown for an additional 16 h.

The cells were harvested by centrifugation at 4500× *g* at 25 °C for 20 min, and then washed with cell-wash-buffer (50 mM Tris-HCl, pH 8.0, 50 mM NaCl). The washed cells were disrupted by sonication with cell-lysis buffer (50 mM Tris-HCl, pH 8.5, 500 mM NaCl, 5% glycerol, 0.25 mM DTT, 25 mM imidazole) in the presence of 100 μg/mL deoxyribonuclease I and 50 μg/mL lysozyme, and the cell debris was then removed by centrifugation at 40,000× *g* for 30 min. The cell lysate was heat-treated at 70 °C for 30 min and centrifuged at 40,000× *g* for 30 min. The supernatant was filtered through a 0.22 μm filter and then loaded onto a Ni-affinity chromatography column (1 mL Histrap™ HP; GE Healthcare, Chicago, IL, USA) equilibrated with buffer A (50 mM Tris-HCl, pH 8.5, 500 mM NaCl, 5% glycerol, 0.25 mM DTT, 25 mM imidazole). The samples were eluted with a gradient of 25–400 mM imidazole in buffer A. The eluted fractions of the proteins were treated with TEV protease and dialyzed against buffer B (50 mM Tris-HCl, pH 8.5, 25 mM NaCl, 5% glycerol, 0.25 mM DTT, 25 mM imidazole) to remove the cleaved His tag. The dialyzed proteins were purified again using the Ni-affinity column, and the flow-through fraction was collected. The proteins were then purified using a 1 mL RESOURCE^TM^ Q column (GE Healthcare) with buffer C (20 mM Tris-HCl, pH 8.5, 25 mM NaCl, 5% glycerol, 0.25 mM DTT). The bound protein was eluted with a linear gradient of NaCl (25–2000 mM). The fractions containing DHS were pooled and then load on to a Superdex 75 16/60 column (GE Healthcare, Chicago, IL, USA) in buffer D (20 mM HEPES, pH 8.0, 200 mM NaCl, 5% glycerol, and 1 mM DTT), and DHS was eluted as a single peak with an estimated molecular weight around 130 kDa, corresponding to the tetramer. Fractions containing purified tetramer were combined and concentrated using VIVAPORE 10/20 (2–20 mL). All steps were monitored by 15% SDS-PAGE.

### 4.2. Crystallization and Data Collection

Preliminary crystallization trials were performed using a crystallization screening kit from QIAGEN (JCSG I-IV, JCSG + and PEG I&II, Hilden, Germany). A crystallization drop containing 1 μL of protein solution and 1 μL reservoir solution was set up and equilibrated against 100 μL of reservoir solution at 20 °C using the sitting-drop method. Initial crystals were grown in a condition containing 100 mM phosphate–citrate pH 4.2, 40% ethanol, 5% PEG 1000. After the optimization of crystallization conditions with respect to pH and precipitant concentration, well diffracted crystals were obtained in a condition containing 100 mM phosphate–citrate pH 4.8, 32% Ethanol, 5% PEG 1000.

For diffraction data collection, the crystals of *Pho*DHS were transferred into a cryo-protectant solution containing 15% glycerol in the reservoir solution, then mounted in a loop and flash-cooled in a stream of nitrogen gas at −173 °C. The X-ray diffraction experiment was performed with a wavelength of 1.0 Å at beamline BL41XU in SPring-8 (Hyogo, Japan). The diffraction data were indexed, integrated, scaled, and merged using the HKL2000 package [30]. The crystals belong to space group P*2_1_2_1_2_1_*, with the unit-cell parameters of *a* = 87.3 Å, *b* = 89.9 Å, *c* = 164.8 Å. Using a calculated molecular weight of 38.9 Da, the Matthews coefficient V_M_ value was estimated to be 2.07 Å^3^Da^−1^ with 4 *Pho*DHS molecules in the asymmetric unit, corresponding to a solvent content of 40.56%. A summary of data collection and processing statistics is given in Table 1.

### 4.3. Structural Determination and Refinement

The structure of *Pho*DHS was solved by molecular replacement with MOLREP [31], using the protein structure of human DHS (PDB ID:1DHS) as a search model [21], which shows 39.5% amino acid sequence identity with *Pho*DHS. Initial rigid body refinement was performed with the REFMAC [32] of the CCP4 program suite [33] and the structure was then rebuilt using the program LAFIRE with CNS automatically [34,35]. Structural refinements were further performed using the program phenix_refine of the phenix program suite [36], followed by manual modifications with Coot [37]. After several refinement cycles, the NAD^+^ molecules were manually built based on 2Fo-Fc and Fo-Fc electron density maps. During the refinement of *Pho*DHS in complex with NAD^+^, the NAD^+^ molecules were rebuilt based on the omit-map several times and the occupancy of NAD^+^ molecules was adjusted manually following check of Fo-Fc and 2Fo-Fc maps. We also rebuilt the binding loop (V288–P299) several times based on the omit-map and the manually fine-tuned residues occupancy. The *R_work_* and *R_free_* factors were monitored during the refinement process, and the latter was calculated from 5% of the reflections. The final structure of *Pho*DHS was refined to a Rwork/Rfree = 17.52/22.95%, respectively. The summary of refinement statistics is shown in Table 1. The atomic coordinates of *Pho*DHS have been deposited in the Protein Data Bank under the accession number 7CMC. The buried areas of inter monomer were calculated using PDBePISA [38].

### 4.4. Analysis of NAD^+^ Binding by UV Absorbance Spectroscopy

The purified *Pho*DHS was mixed with NAD^+^ (Roche, Nutley, New Jersey, USA) at the molar ratio of 1:30. The complex of *Pho*DHS and NAD^+^ was dialyzed in buffer C (20 mM HEPES, pH 8.0, 200 mM NaCl, 5% glycerol, and 1 mM DTT) by using EasySep (TOMY, Tokyo, Japan) to remove free, excess NAD^+^. The final concentration of the *Pho*DHS tetramer was about 5.87 μM. The UV absorption spectra of the dialyzed samples were measured by using a DU800 Spectrophotometer (Beckman, Brea, California, USA) at the scanning speed of 1200 nm/min. The dialyzed buffer was taken as the blank. NAD^+^ at 106 μM in buffer D (20 mM HEPES, pH 8.0, 200 mM NaCl, 5% glycerol, and 1 mM DTT) was used as control. The absorption curve of 2*NAD^+^ (11.74 μM), a 1:2 molar ratio of *Pho*DHS:NAD^+^ was similarly measured using the control curve of NAD^+^ (106 μM) as the reference.

## Figures and Tables

**Figure 1 ijms-21-05509-f001:**
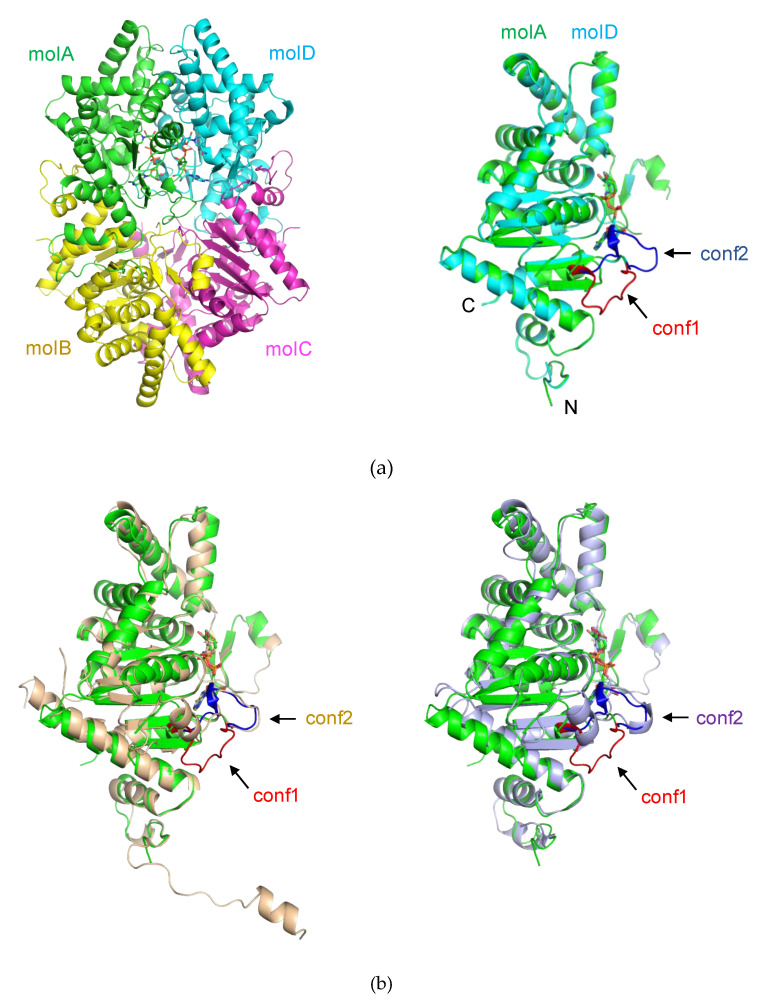
Crystal structure of deoxyhypusine synthase (DHS). The protein and NAD^+^ is shown by the ribbon and stick, respectively. (**a**) Overall structure of *Pyrococcus horikoshii OT3* DHS (*Pho*DHS): (left) A tetramer of *Pho*DHS in an asymmetric unit with monomeric units colored green (molA), yellow (molB), magenta (molC) and cyan (molD). (right) Superimposition of molD (cyan) onto molA (green). A loop (V288–P299) around the NAD^+^ binding site is observed in two different conformations (blue and red) in molA (conformation 1, conf1) and molD (conformation 2, conf2), respectively. While the conf1 in molA is away from NAD^+^, the conf2 in molD moves towards NAD^+^. (**b**) Structural superposition of *Pho*DHS (green) with human (wheat, PDBID:1DHS) and *Trypanosoma brucei* (light purple, PDBID:6DFT) DHS. The conf1 and conf2 loop in the comparison molecules are labeled (in their respective colors), and the loop in conf1 (red) of molA is added for comparison.

**Figure 2 ijms-21-05509-f002:**
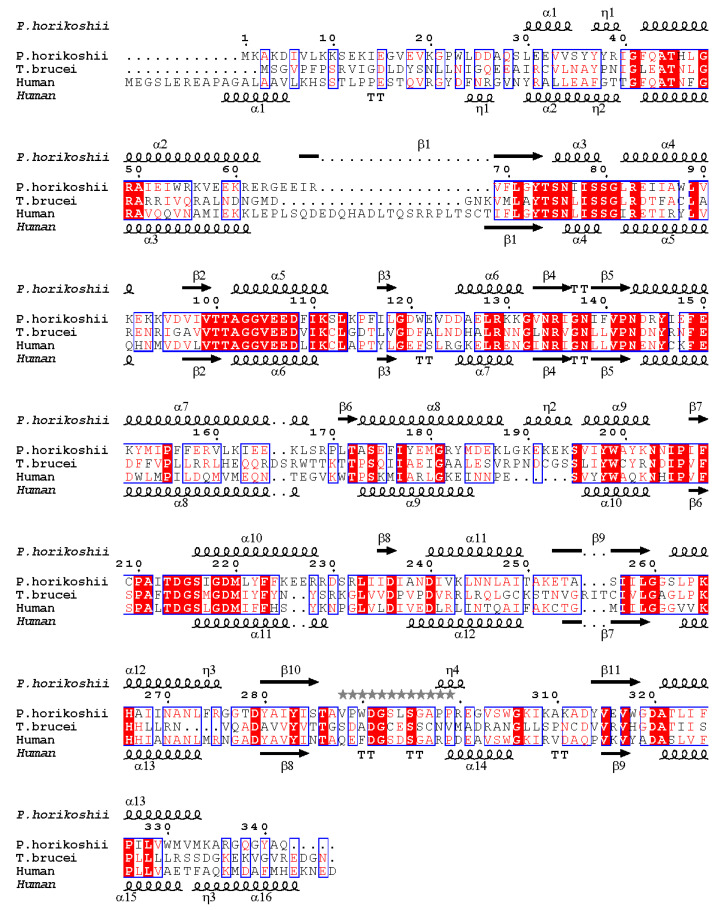
Sequence alignment of *Pho*DHS with DHS from human and *T. brucei*. The ESPript 3.0 server was used to output the alignment [28]. The blue box above the sequence indicates that the sequences of at least two out of three organisms share a similarity, while the sequences identical in all the three organisms are marked with red fills. The secondary structure shown by the respective crystal structures for *Pho*DHS and human DHS are labeled at the top and bottom, respectively. The loop 288–299 with alternative conformations is indicated with grey stars above.

**Figure 3 ijms-21-05509-f003:**
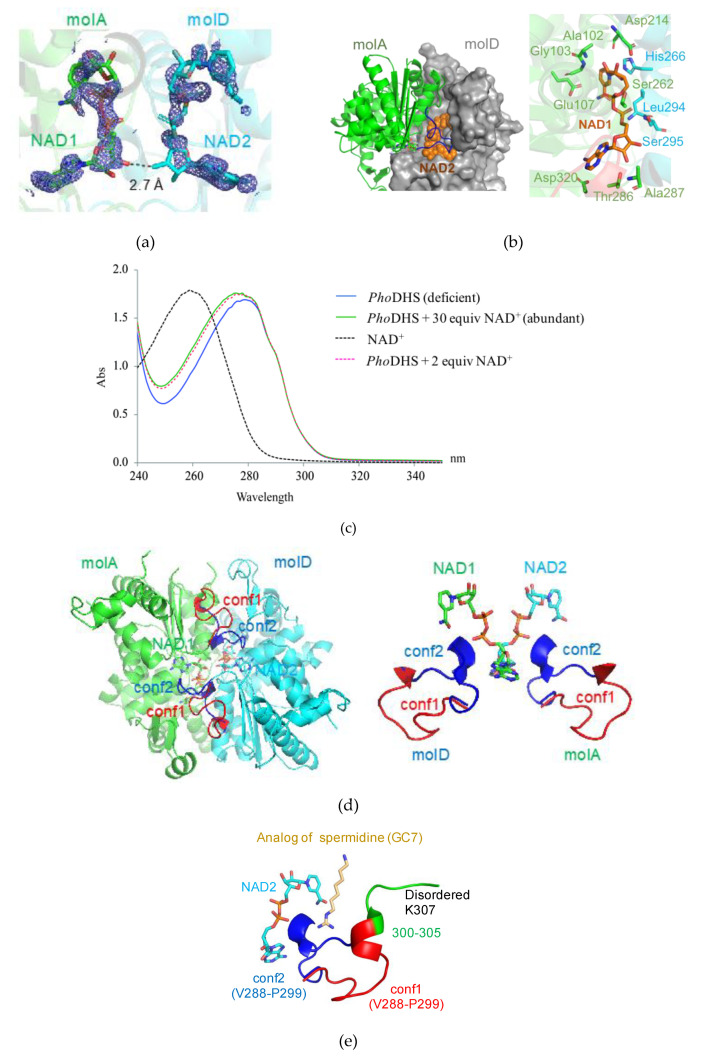
Active site of *Pho*DHS-bound NAD^+^. (**a**) Electron density map for NAD^+^. NAD^+^ is superposed with the 2Fo-Fc electron density map contoured at the 1σ level (blue). (**b**) Binding geometry of NAD^+^ by *Pho*DHS. (left) NAD^+^ is bounded to the interface of the dimer in a sandwich manner. (right) NAD^+^ extensively interacts with residues from both monomers via a hydrogen bond and electronic interaction. (**c**) Quantification of NAD^+^ bound to *Pho*DHS in solution by UV absorbance spectroscopy. An obvious difference in the absorption curves for *Pho*DHS alone (no NAD^+^ added) and supplemented with NAD^+^ as *Pho*DHS + 30 equiv NAD^+^ (molar ratio *Pho*DHS:NAD of 1:30), shows the binding of NAD^+^ to vacant sites in purified *Pho*DHS. The absorption curve of *Pho*DHS + two equiv NAD^+^ (molar ratio 1:2) correlates well to the curve *Pho*DHS + 30 equiv NAD^+^, indicating that *Pho*DHS could only accommodate two additional NAD^+^ even if excessive NAD^+^ molecules are available. (**d**) (left) Closed (blue) and open (red) forms of the binding loop V288–P299 with NAD^+^ bound. (right) Alternative conformations of the loop involved in NAD^+^ and spermidine binding. (**e**) Superposition of GC7 onto *Pho*DHS. GC7 is superposed onto *Pho*DHS by structural alignment with human DHS in complex with GC7. The loop of two conformations (conf1 and conf2) in molA and molD was generated by the superposition of the two monomers.

**Figure 4 ijms-21-05509-f004:**
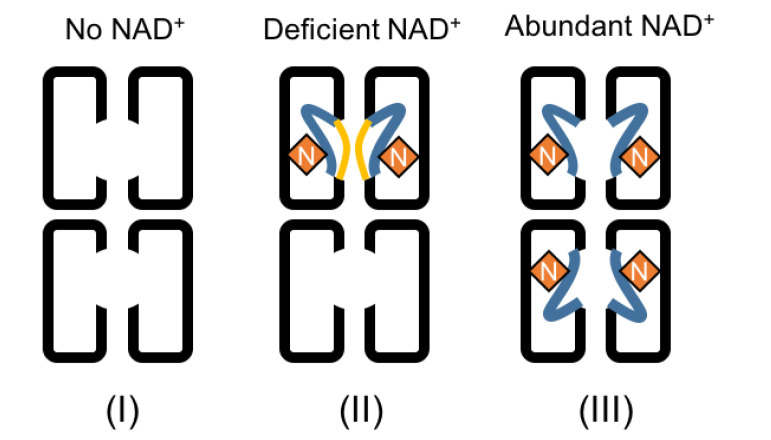
A model of the conformational dynamics in DHS, regulated by NAD^+^. Three binding conformations are shown. On the left, the spermidine binding loop is disordered when there is no NAD^+^ binding to DHS. In the middle, the loop is partially stabilized and structured in either conf1 or conf2 when NAD^+^ binds to DHS. The binding of one NAD^+^ to DHS could promote a second NAD^+^ to bind to another monomer of the dimer, and binding of both NAD^+^s would be stabilized via the formation of the hydrogen bond between them. The spermidine binding loop is still disordered in another dimer of the tetramer which remains NAD^+^-free. On the right, the spermidine binding loop of all four monomers of *Pho*DHS will be ordered in conf2 in the presence of abundant NAD^+^, based on the current structure and human DHS structure. Yellow lines: loop in conf1 away from NAD^+^; blue lines: loop in conf2 towards NAD^+^; N: NAD^+^ molecules.

**Table 1 ijms-21-05509-t001:** Summary of data collection and refinement statistics.

**Data Collection**	
Crystal	Native DHS
Space group	*P*2_1_2_1_2_1_
Beamline	Spring 8 BL41XU
Wavelength (Å)	1.0000
Unit-cell parameter	a = 87.3 Å, b = 89.9 Å, c = 164.8 Å, α = β= γ = 90°
Resolution range (Å)	50–2.20 (2.28–2.20)
Total number of reflections	666,150 (6550)
Completeness (%)	100.0 (100.0)
Redundancy	7.4 (7.4)
I/σ(I)	29.1 (5.2)
R-merge	0.110 (0.479)
**Refinement Statistics**	
Resolution range (Å)	34.42–2.20
R_work_/R_free_ (%)	17.52/22.95
R.m.s.d bond lengths (Å)	0.07
R.m.s.d angles (°)	0.93
No. non-H atoms	
Proteins	9817
Water molecules	408
NAD^+^ molecules	88
Average B value	
Protein	35.17
Water molecules	41.81
NAD^+^ molecules	71.98
Ramachandran Plot (%)	
Favored	98.91
Allowed	1.09
Outline	0.00

Values in parentheses are for the highest resolution shell. Rwork = Σhkl ||Fobs|−|Fcalc||/Σhkl |Fobs|, Rfree was calculated for 5% randomly selected test sets that were not used in the refinement.

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
