# Peer review of "Flexible NAD+ Binding in Deoxyhypusine Synthase Reflects the Dynamic Hypusine Modification of Translation Factor IF5A"

_ijms, 2020, doi:10.3390/ijms21155509_

Round 1
Reviewer 1 Report
The manuscript by Chen et al. describes the crystal structure of an archaeal deoxyhypusine synthase. Like previously studied prokaryotic DHS, the enzyme is tetrameric and composed of a dimer of tightly associated dimers. The authors observe that two subunits bind NAD+ whereas the two others are in the apo form. It is shown that the two NAD+ molecules are present in solution in the purified enzyme and that the enzyme can accomodate two additional NAD+. Interestingly, the V288-K311 loop is disordered in the absence of NAD+ whereas it can adopt two alternative conformations in its presence. These data suggest how the activity of DHS may be controlled by the availability of NAD+. This is an interesting study, adequately performed and clearly reported. The following points should however be addressed:
1- Table 1 was not included in the available manuscript.
2- The full name of the studied enzyme should be indicated in the title instead of simply DHS.
3- An analysis of the packing interactions and of their possible influence on the conformation of the V288-K311 loop should be included.
4- A Figure showing parts of the final electron density, in particular for the two conformations of the V288-K311 loop should be added.
Reviewer 2 Report
The manuscript by Chen et al reports first crystal structure from Pyrococcus horikoshii. Strikingly the structure shows some conformational changes in relation to the previously reported DHS structures (eg human). Despite some novelty and proposed mechanism of NAD concentration dependent fine-tuning of translation it raises major concerns as described below.
Despite of relating to Table 1, the Table 1 does not exist in the manuscript.
Crystallographic description is lacking crucial information e.g. space group.
The statistics reported in the released structure significantly differ from those reported in the manuscript!!
The careful inspection of the deposited pdb revealed that:
- modeling of two conformations of the flexible loop may be wishful thinking due to the lack of strong density. I do not doubt that the loop is flexible but, one of the conformations has no sufficient density
- it seems that the deposition authors did not refine occupancies of the loop residues (compensated by B-factors)
- electron density for one of the NAD molecules is too weak to model the molecule at high occupancy
Experiments that should demonstrate NAD stoichometry are not sufficient. It should be determined by eg. HPLC
Also:
35 - it should be translation not translation initiation factor (old nomenclature)
57 - K329 is human, indicate
64 - dimer of a tight dimer - very vague
107 - no space group
139 - poor density for loops
In conclusion, I do not endorse publication of the submitted manuscirpt in IJMS journal. I strongly encourage the authors to revise their work and submit to more specialized, crystallographic, journal.
Reviewer 3 Report
This manuscript is a short report of a novel structure of Pyrococcus horikoshii OT3 deoxyhypusine synthase (PhoDHS). Overall, there are similarities in the fold and assembly of PhoDHS compared to known structures of the human enzyme. This report describes two new findings for this type of enzyme which are the main points of the report: preferential occupancy of two of the four active sites by NAD+ and a "lid" mechanism of an active-site adjacent loop observed in both open and closed conformations. In principle, the report is scientifically sound; however, there are two major concerns, detailed below, related to each of these new findings with this structure of PhoDHS that must be thoroughly addressed. As deposited (accessed 6/25/2020), PDBID 4P63 (including model, electron density and validation report) does not beyond reasonable doubt support the main points of the report.
Major Concerns:
Preferential NAD+ occupancy: The NAD+ ligands modeled in chains A and D of the model, as deposited under entry 4P63, likely are not present at 100% occupancy as these ligands were refined. Average heavy atom B-values for the ligands is 91 compared with 38 for the protein chain and 46 for the solvent. This model should be re-refined with partial occupancy of NAD+ for both chains and the results of such refinement compared with the 100% occupancy refinement to decide which refinement approach yields a model that best represents the electron density.
Binding loop "lid": The "conformation 2" of the binding loop is first reported here. Tracing through the model and electron density (as retreieved from the PDB) does show reasonable backbone connectivity for conformation 2 in chain D. The density is less convincing for chain A. However, the PDB validation report (as retreived from the PDB) lists conformation 2 residues 290-297 of chains A and D as real space R outliers with RSRZ > 3.0 (exceeding the >2.0 threshold for significance). This analysis suggests that these residues do not, in reality, correspond to the observed electron density, and eight of these residues (either from chains A or D) participate in interatomic clashes also flagged in the validation report. Furthermore, the validation reports 7.9% of residues being real space outliers (RSRZ >2.0). Can these validation "warning signs" be remedied or explained?
Minor Issues:
Section 2.1: How many (what percentage of) residues are observed/modeled for each molecule of the crystallized amino acid sequence? (Information included in Table 1 or Section 4.3, annotated onto Figure 2 or described in Supplementary Materials?)
Line 92: Please provide a database reference (Uniprot, NCBI, etc.) for the PhoDHS sequence, if such exists.
Lines 117-118: The authors should not say whether or not the N-terminal two-turn alpha helix region of PhoDHS "obstruct[s] the active site" as the residues in question are not observed in the electron density maps. All that can be concluded is that this region of the model does not reveal a conformation in which the active site is blocked. It is likely that disorder of this region contributes to the missing electron density, of course, but observed as obstructing in PDB 1DHS versus not observed, possibly due to disorder, is as far as this PhoDHS model confirms. Further, the model only suggests that the active site may be accessible, not confirms that the active sites "are accessible" as the text currently states.
Lines 135-138: Is the dissociation constant (Kd) known for NAD+ in this binding mode of PhoDHS? If so, is the constant sufficiently small to support the argument that PhoDHS bound NAD+ from the expression host and this binding was stable during purification and crystallization? What is the modeled NAD+ occupancy ratio and B-factor (and how does the NAD+ B-factor compare with the B-factors of side-chains making direct contact to the ligand)? Was it explored as to whether or molecules (citrate, phosphate, PEG1000) from the crystallization condition could satisfy the "ligand" electron density observed? Were omit maps calculated for the NAD+ ligand? Such results could be included in Supplementary Materials. The UV-Vis experiment described in Lines 174-181 does strengthen the argument that PhoDHS tetramer carried over two molecules of NAD+ from expression.
Lines 141-142: Would it be possible to show electron densities for the two conformations observed for the binding loop? Such results could be included in Supplementary Materials.
Lines 174-181: UV-Vis experiment: is there a known means of obtaining truly apo-PhoDHS with no NAD+ bound? If so, comparing the UV-Vis spectra of this species with those measured in this experiment and shown in Figure 3 would further strengthen the argument of having sites that are preferentially satisified under deficient [NAD+] compared to abundant [NAD+].
Line 177: please make the following changes: "PhoDHS + 30*NAD+" to "PhoDHS + 30 equiv NAD+"; "PhoDHS + 2*NAD+" to "PhoDHS + 2 equiv NAD+". Please also make these changes in Figure 3(c).
Figure 1: Please give both the relative and absolute contouring of the electron dnesities shown in panel A. I downloaded the coordinates and map coefficients from PDBID 4P63, and, when contoured at 1sigma (0.2744e/Å^3) in coot, the electron densities do not appear to be as complete as they are shown in Fig 1 panel (a).
Figure 2: Unexplained and/or mis-rendered characters on the sequence annotation diagram. The boxes, above PhoDHS residues 32-33, for example, appear to be mis-rendered characters. The stars above PhoDHS resides 288-299 do not have an explanation given in the Figure 2 Legend though one assumes they are there to denote the binding loop. In the Figure 2 legend, please cite EndScript, ESpript or whichever similar tool was used to generate the annotated sequence alignment.
Figure 3: It might be helpful to the reader to add to the legend displayed in panel (c) so that it reads "PhoDHS (deficient)" and "PhoDHS + 30 equiv NAD+" (abundant). If a truly NAD+ free spectrum could also be obtained for PhoDHS, please include it here labeled as "PhoDHS (apo)" or "PhoDHS (no NAD+)"
Round 2
Reviewer 2 Report
The provided resubmitted manuscript by Chen et al. has been improved. However, it still need introduction of major changes.
- The authors claim that they re-refined the model and submitted it to the PDB. Please provide new PDB code (old one should become obsolete). Please provide the PDB validation report of the re-refined structure.
- Line 26:
“Our results demonstrate the dynamic structure of PhoDHS at NAD+ and spermidine binding site, with conformational changes that are likely to respond to the local concentration of NAD+ and thus fine-tune the regulation of the translation process via hypusine modification of IF5A.”
I suggest to tune-down this statement and writing : …that may be the response to the local NAD+ concentration, and thus.. - In Table 1, please provide CC1/2 value (for all and high res shell). (Suggestion of resolution cut-off at CC1/2 ~ 0.5 for high-res shell according to Karplus and Diederichs Science 2012)
- Inconsistency in the reported R values:
176/0227 – table 1
0.166/0.203 – PDB deposit 4P63 (which should become obsolete and replaced by new deposition with new values and PDB code)
0.19/0.23 – Line 363 in methods section.
This is unacceptable! Please: 1. provide new PDB code, 2. provide new statistics 3. Keep the statistics the same everywhere.
Minor:
Line 361 – re-write the sentence, remove “About the binding loop of V288–P299..”
I suggest Also, we rebuilt the binding loop (V288-P299) basing on the Fo-Fc omit-map and the manually fine-tuned residues occupancy.
Line 356 – phenix not phoenix, please check it throughout the manuscript
Author Response
"Please see the attachment

Reviewer 3 Report
The authors have adequately addressed my concerns with this manuscript with the exception that the PDB validation report is a preliminary report that does not confirm that updated coordinates and structure factors have been deposited with the PDB for entry 4P63. Once the authors can provide to the journal a PDB deposition validation report confirming updates to the 4P63 entry in the PDB, I recommend this article for publication.
Round 3
Reviewer 2 Report
The manuscript has been improved and all of my concerns were addressed.